# Implementation of Mitigation Measures to Reduce Phosphorus Losses: The Vestre Vansjø Pilot Catchment

**Marianne Bechmann \*, Inga Greipsland and Anne Falk Øgaard**

NIBIO—Norwegian Institute for Bioeconomy Reseach, P.O. Box 115, 1431 Aas, Norway; inga.greipsland@nibio.no (I.G.); anne.falk.ogaard@nibio.no (A.F.Ø.)

\* Correspondence: marianne.bechmann@nibio.no; Tel.: +47-4121-9506

**Abstract:** Diffuse phosphorus loss from agricultural fields is an important contributor to the eutrophication of waterbodies. The objective of this study was to evaluate a pilot project for the implementation of mitigation measures to reduce P losses. The pilot project is situated in southwestern Norway and, covers a 14-year period (2004–2018). It included data on the implementation of mitigation measures and water quality monitoring for six small catchments. The mitigation measures consisted of no tillage in autumn, reduced P fertilizer application, grassed buffer zones, and sedimentation ponds. Extra efforts were made to reduce diffuse P losses during the period from 2008 to 2010. The project comprised economic incentives, an information campaign, and farm visits. Data from 2004 and 2010 showed that the use of P fertilizer during this period decreased by 80% and the area of no-till in autumn increased in all six catchments and covered 100% of the area in three of the six catchments in 2010. However, with decreased economic incentives after 2010, the degree to which the mitigation measures were implemented was reversed; P-fertilization increased, and no-till in autumn decreased. No significant effects of mitigation measures on total P and suspended sediment concentrations were detected. We conclude that economic incentives are necessary for the comprehensive implementation of mitigation measures and but that it is not always possible to show the effect on water quality.

**Keywords:** agriculture; erosion; fertilizer; buffer; sedimentation pond; sewage; suspended sediments; water quality; Water Framework Directive

## 1. Introduction

Eutrophication of waterbodies in catchments with a high share of agricultural land is common. Both nitrogen and phosphorus contribute to this, but P has been shown to be the limiting factor for eutrophication in most freshwaters [1]. During the last decades, there has been focus on removing point sources such as waste water [2]. After removing the main point sources of nutrients, agricultural non-point sources are often pointed out as main contributors of P to surface waters. Major efforts have been initiated to reduce the problems of eutrophication [3].

Agricultural mitigation measures aimed at reducing non-point sources of P include better nutrient management, improved soil and crop management, and increased retention in the landscape [3,4]. Measures are implemented at different scales: the plot scale (e.g., soil tillage), hill slope scale (e.g., grassed buffer strips) or at the headwater catchment scale (e.g., constructed wetlands). The effect of various measures on suspended solids (SS) and total phosphorus (TP) has been documented at the specific scales [3,5]. The effect of soil tillage methods was documented by Reference [6], who showed that no-till in autumn reduced soil losses and TP by on average 80%. The effect of reduced P

application and soil P status has been documented by several authors, e.g., Reference [7]. Furthermore, the effect of grassed buffers and sedimentation ponds were documented for Norwegian conditions by References [8,9], respectively. However, only a few studies have been able to demonstrate the effect of mitigation methods at the catchment scale [1,9]. Haygarth et al. [10] carried out intensive monitoring of two catchments and found that reducing P application in the form of mineral fertilizer or manure had no effects on P losses from the catchments. However, they showed indications of the effect of soil tillage on sediment transport.

Many pilot projects aimed at improving water quality in catchments affected by agriculture have been initiated in Norway under the umbrella of the Water Framework Directive (WFD) [11]. The pilot projects introduced packages of agricultural mitigation measures designed to improve water quality in surface water and groundwater aquifers [12]. The pilot projects often subsidies farmers for their participation, since implementing mitigation measures can often cause a decrease in production and thereby lead to loss of income [13,14]. To increase the farmers' participation in mitigating agricultural non-point source P losses, it is recommended that stakeholders are involved from the onset of the project [14]. Furthermore, operational monitoring of water quality should be conducted with a view to documenting changes in water quality [15].

There is limited knowledge about whether these pilot projects meet their objectives of implementing mitigation measures and improving water quality of the recipient. This paper summarizes the outcome of a pilot project intended to improve the water quality of a eutrophic lake, the lake Vestre Vansjø in Norway. The objective was to evaluate (a) the effect of the pilot project on the implementation of mitigation measures, and (b) the effect of mitigation measures on water quality.

## 2. Materials and Methods

### 2.1. Case Study Area

The Vestre Vansjø catchment in southeastern Norway was described and initially evaluated by Bechmann and Øgaard [16]. The six small catchments were chosen to be representative of the agricultural areas contributing nutrients to the lake Vestre Vansjø (Figure 1). The catchments of these six streams range from 13 to 478 ha in size, and the agricultural area constitutes 12 to 90% of the catchment areas (Table 1). Three of the catchments are dominated by clay loam soils with cereal production, whereas the three other catchments are dominated by sandy loam soils with the production of potatoes and vegetables in addition to cereals. Most agricultural areas are artificially tile drained. In Norway, erosion risk at standard autumn-ploughing is available from national erosion risk maps and are classified as low (erosion risk class 1: <500 kg soil ha$^{-1}$), medium (erosion risk class 2: 500–2000 kg soil ha$^{-1}$), high (erosion risk class 3: 2000–8000 kg soil ha$^{-1}$) or very high (erosion risk class 4: >8000 kg soil ha$^{-1}$) [17]. The erosion risk of the agricultural soils in the six catchments is mainly low to medium, with 27 to 83% in erosion risk class 1 and none of the areas in erosion risk class 4 (>8000 kg soil ha$^{-1}$) (Table 1). The low erosion risk relates to the gentle slopes in these areas. Animal production consisted of swine production in Augerød and chicken production in Guthus, but this production ended during the first years of the pilot project. The average annual precipitation in the area is 829 mm (1961–1990; [18]). The six small case study catchments were monitored for water quality throughout the 14-year study period.



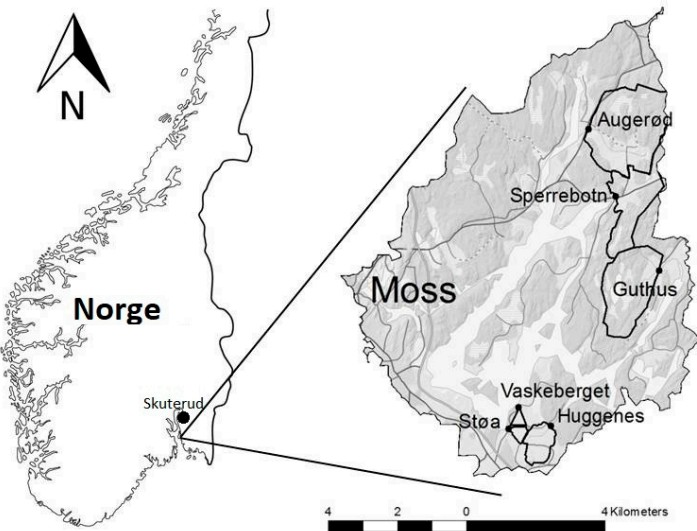

**Figure 1.** Location of the Vestre Vansjø catchment with the six case study catchments.

**Table 1.** Total catchment area, percent agricultural area, production, soils and erosion risk (ER, % of agricultural land) [17].

| Catchment | Total Area ha | Agriculture % | Production | Soils | Erosion Risk ** (% of Agricultural Land) | | |
|---|---|---|---|---|---|---|---|
| | | | | | ER-cl. 1 | ER-cl. 2 | ER-cl. 3 |
| Augerød *** | 478 | 20 | Cereals, swine * | Clay loam and silt loam | 40 | 58 | 2 |
| Guthus *** | 315 | 12 | Cereals, chicken | Clay, silty loam and org. | 65 | 35 | 0 |
| Sperrebotn | 248 | 19 | Cereals | Silt loam and clay loam | 27 | 73 | 0 |
| Støa | 16 | 89 | Cereals, vegetables, potato, grass | Sandy loam, and silt loam | 83 | 17 | 0 |
| Vaskeberget *** | 13 | 91 | Cereals, vegetables, potato | Sandy, silty and clay loam | 45 | 55 | 0 |
| Huggenes | 81 | 85 | Cereals, vegetables, potato | Sandy, silty and clay loam | 57 | 42 | 1 |

* No swine after 2006, ** at autumn ploughing; Low ER-class 1: <500 kg soil ha$^{-1}$, medium ER-class 2: 500–2000 kg soil ha$^{-1}$, high ER-class 3: 2000–8000 kg soil ha$^{-1}$ and very high ER-class 4: >8000 kg soil ha$^{-1}$; *** Only cereals after 2011.

### 2.2. The Lake Vestre Vansjø Project

The Vestre Vansjø catchment is part of the Morsa pilot project under the WFD and the monitoring of water quality started in small streams in the Vestre Vansjø catchment in 2004 (Table 2). Both point and non-point sources of P were targeted during the Morsa pilot project [16] through an action plan. In addition, the Norwegian Ministry of Food and Agriculture from 2008 to 2010 funded a mitigation project in the Vestre Vansjø catchment (Table 2) during which farmers were encouraged to sign environmental contracts that included receiving subsidies to cover extra costs and potential loss of income. These contracts included:

- Using less P fertilizer than the national recommended level
- No soil cultivation during autumn
- No cultivation of potatoes or vegetables on fields frequently prone to flooding
- Establishment of eight-meter-wide grassed buffer strips along open water
- Establishment of grassed waterways where there is a large risk of erosion
- Establishment of constructed wetlands where recommended

**Table 2.** The timeline of the pilot project, mitigation project and regulations in the Vestre Vansjø catchment.

| 2004 | 2005 | 2006 | 2007 | 2008 | 2009 | 2010 | 2011 | 2012 | 2013 | 2014 | 2014 | 2015 | 2016 |
|------|------|------|------|------|------|------|------|------|------|------|------|------|------|
| | | | | | | | Water Framework Pilot project | | | | | | |
| | | | | Mitigation project with environmental contracts | | | Environmental contracts by County Governor | | | | | | |
| | | | | | General regulation, the 40/60 regulation and subsidies in all ER *-classes | | | | Removal of 40/60 regulation; no subsidies in ER *-class 1 | | | | |

* ER = erosion risk.

An agricultural advisor was responsible for the dialogue with farmers to encourage them to sign contracts. Farm visits, information campaigns, and dialogue meetings with farmers were important during the mitigation project (2008–2010). The project focused on cooperation between local authorities (e.g., the county governor), agricultural advisors, farmers, and research scientists to reduce P losses from agricultural areas.

At the start of the mitigation project in spring 2008, 29 of the 40 farmers signed environmental contracts. The share of the agricultural area in the six small catchments that had contracts during the period 2008 to 2010 was: Guthus 100%, Sperrebotn 92%, Augerød 75%, Støa 70%, Vaskeberget 0%, and Huggenes 46%. Farmers with intensive vegetable production (Støa, Vaskeberget, and Huggenes) were less willing to participate in the mitigation project than farmers producing cereals (Augerød, Guthus, and Sperrebotn) [16]. However, the farmers without contracts also provided information on soil practices and P application from 2011–2016.

After the first three-year period (2008–2010), a similar approach with environmental contracts was funded by the county governor of Østfold including all six catchments from 2011 to 2013. During this period (2011–2013), however, there was no one responsible for the dialogue with farmers and not as much focus on cooperation between institutions. In addition to the contracts, there was a general regulation in the region during the period 2009 to 2012 that at least 60% of the agricultural area on each farm should not be tilled in autumn. Furthermore, all areas were granted subsidies for no-till in autumn until 2012. The regulations changed from 2013 and no-till in autumn was no longer required for 60% of the area on each farm and subsidies were stopped for areas in erosion risk class 1 (<500 kg soil loss ha$^{-1}$). These new regulations indicated that the authorities wanted to target mitigation measures to high-risk areas and increase focus on agricultural production.

During the mitigation project period, the farming activities were documented. Data on P application to agricultural areas were collected from the farmers who had signed contracts whereas data on soil tillage, buffer zones, and sedimentation ponds were based on statistical data and are complete for all years (Norwegian statistics). Data on P application to agricultural areas are only complete for Guthus, and data for some of the years are missing for the other catchments. Information on agricultural management was collected for each year between 2011 and 2017, in the form of a questionnaire sent to farmers in 2017. In areas where the information was lacking, the distribution of agricultural management was scaled up from a minor area to the whole catchment.

Information about sewage treatment systems for private households was obtained from the municipalities (Våler and Rygge).

## 2.3. Water Quality Sampling and Runoff Estimation

The in-stream monitoring of water quality was operational monitoring carried out to assess any changes in the status resulting from the Programme of Measures (WFD). The number of samples, monitored parameters and periods are shown in Table 3.

**Table 3.** The variation in annual number of water samples for total phosphorus (TP) and suspended sediment (SS) from 1 May to 30 April in 2004 to 2018 for the streams of the six catchments sampled. Annual runoff (mm) measured in the Skuterud catchment.

| | Agrohydrological Year (1 May to 30 April) | | | | | | | | | | | | | |
| --- | --- | --- | --- | --- | --- | --- | --- | --- | --- | --- | --- | --- | --- | --- |
| | 04/05 | 05/06 | 06/07 | 07/08 | 08/09 | 09/10 | 10/11 | 11/12 | 12/13 | 13/14 | 14/15 | 15/16 | 16/17 | 17/18 |
| No of samples (TP) | 10–11 | 14 | 22–27 | 22–26 | 19–27 | 16–21 | 7–17 | 9–25 | 5–18 | 7–18 | 6–17 | 7–17 | 6–11 | 16–23 |
| No of samples (SS) | - | - | 21–26 | 22–26 | 19–27 | 16–21 | 7–17 | 9–25 | 5–18 | 7–18 | 6–17 | 7–17 | 6–11 | 16–23 |
| Runoff (mm) | 331 | 448 | 745 | 737 | 631 | 518 | 512 | 652 | 535 | 761 | 749 | 658 | 288 | 641 |

The six small streams were monitored by manual water sampling carried out bi-weekly. Samples were transported directly to the laboratory for cooling and were analyzed the same or the following day. Data on concentrations are presented from 1 May to 30 April each year to evaluate the effect of one growing season on the water quality the following year. Extra samples were taken during storm events. These results are excluded from the evaluation of trends and the comparison between years. The annual number of samples from each of the streams varied from 5 to 27 for both TP and SS. The low number of samples for some streams in certain years was caused by long periods of dry or frozen conditions (Table 3).

There was no runoff data on the monitored catchments and, therefore, data from the Skuterud stream (about 30 km north of the Vestre Vansjø catchment) was used (Table 3). The Skuterud catchment (470 ha) is dominated by agricultural land, and discharge is continuously measured by a crump weir, with automatic registration of half-hourly water level using a Campbell Scientific data logger [19]. The annual flow-weighted concentrations were calculated as the annual loads divided by annual flow. Annual loads (not shown) were calculated by linear interpolation of concentrations in single samples.

*2.4. Chemical Water Analyses*

All water samples were analyzed for the concentration of TP. From 2006 and onwards samples were also analyzed for the concentration of suspended sediments (SS). Some samples were also analyzed for the concentration of dissolved reactive phosphorus (DRP).

Unfiltered samples were used to determine TP by digestion with $K_2S_2O_8$, and filtered samples (<45 μm) were analyzed for DRP. Phosphorus in all filtrates and neutralized digests was analyzed spectrophotometrically using Murphy and Riley's ammonium molybdate blue method [20]. Suspended sediments were determined by filtering an exact sample volume of 25 to 250 mL after thorough mixing (containing at least 5 mg SS) through a pre-weighed fiberglass filter (Whatman GF/A). One water sample each month was analyzed for Termostable Coliform Bacteria (TKB).

*2.5. Soil Sampling*

The soil P status of agricultural soils in the six catchments was based on soil analyses carried out for the purpose of planning the farmers' nutrient management. The soil samples were analyzed using the ammonia lactate extraction method (P-AL) [21], where the soil was extracted for 1.5 h with a solution of 0.1 M ammonium lactate and 0.4 M acetic acid in the ratio of soil to a solution of 1:20 (*w/v*). Results at the field level were provided by the farmers themselves or the local authorities. For the period 2006 to 2011, 142 samples were included in this study, and for the period 2016 to 2017, 182 samples were included.

*2.6. Statistical Analyses and Calculations*

A paired sample *t*-test was used to determine if any significant changes in the soil P status had occurred between the two periods 2006 to 2011 and 2016 to 2017. To evaluate trends in average TP and

suspended sediment concentrations both linear regression analysis and the Mann–Kendall test were used [22].

Soil management in the catchments is presented as a percentage of the agricultural area. A soil tillage indicator was calculated for each catchment and year. The tillage factors are based on Reference [6]. A factor for turf-grass could not be estimated due to lack of knowledge on the effect of turf-grass on erosion at the plot scale. An average soil tillage indicator was calculated for the five catchments with no turf-production and year as follows:

$$\text{Soil tillage indicator} = \% \text{ grassland} \times 0.05 + \% \text{ no-till in autumn} \times 0.2$$
$$+ \% \text{ autumn harrowed} \times 0.5 + 1 \times \% \text{ autumn till/vegetable/potato.}$$

Regression analyses were carried out for fixed variables (size of catchment, share of agricultural area, erosion risk, soil P status, and average P application) for the six catchments and for annual variables (soil tillage, P application, sewage, grassed buffer zones, sedimentation ponds, precipitation amount, precipitation intensity) for each catchment. The statistical program Minitab version 18.1 (Minitab, State College, PA, USA) was used for these statistical analyses.

## 3. Results and Discussion

### 3.1. Implementation of Mitigation Measures

In the catchments Augerød, Guthus, and Sperrebotn, soil tillage in autumn was reduced considerably after the start of the mitigation project (2008) compared to the status in 2004. Autumn tillage was zero or close to zero when the contracts were implemented (Figure 2). In Sperrebotn, no-till in autumn dominated the cereal areas in all of the years. But in Augerød and Guthus from 2011 to 2016 soil tillage in autumn increased to roughly the same level as in 2004. In Støa, Vaskeberget, and Huggenes there were fewer areas with no-till in autumn compared to the first three catchments. The Vaskeberget catchment area was dominated by autumn ploughing in all of the years.

Phosphorus application in 2004 reflects the common fertilization practice before more attention was focused on reducing application (Table 4). For the contracted area, the total annual P application was reduced in total by 12 tons from 2004 to 2010, corresponding to an 80% reduction. After the end of the mitigation project period, the fertilization levels increased in most catchments, though the average fertilization level was still lower in 2016 than at the start of the pilot project period. The lowest P application rates were registered during 2008 to 2013, with no P application at all in Augerød and Støa in certain years (Table 4). In 2004, the average P application in the catchments from which we have obtained data from the farmers varied from 18 to 28 kg P ha$^{-1}$ (Table 4). The highest average P application was found in the catchment Huggenes, which produces vegetables in combination with cereals. This is explained by a generally higher P requirement in vegetable production and by the large economic benefit of producing high yields of these crops [23]. In our study, the share of vegetable crop area was not constant over the years studied, and the annual variation in the P application rate for each catchment may also be related to variation in crops. The zero P application in Støa in 2008 to 2009 for turf production resulted from contract requirements. Some farmers even chose not to grow onion in the Vestre Vansjø catchment but rather moved this production to areas outside the catchment due to the restrictions on P application.

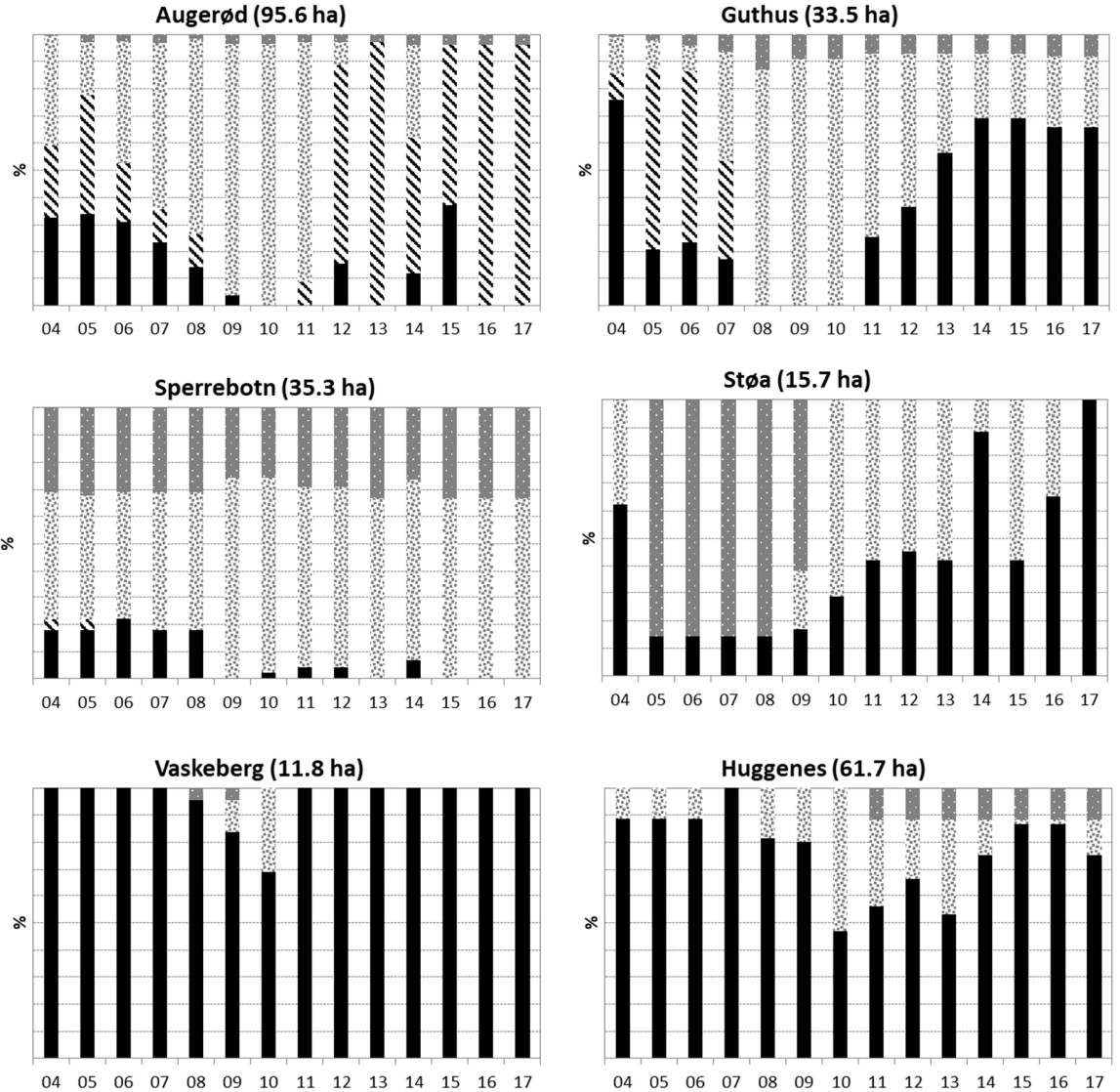

**Figure 2.** Soil management in the six catchments in the years 2004 to 2010. Arable land (ha) in the catchments are shown in brackets, and years are on the *x*-axis. Grass includes all grass-covered areas.

**Table 4.** Average [1] P fertilization [2] for farms with contracts in the different catchments. nd = no data.

| Catchment | 2004 | 2007 | 2008 | 2009 | 2010 | 2011 | 2012 | 2013 | 2014 | 2015 | 2016 |
|---|---|---|---|---|---|---|---|---|---|---|---|
| | | | | | | kg P/ha | | | | | |
| Augerød | 18 | 10 | 7 | 0 | 7 | 7 | 10 | 12 | 13 | 17 | 8 |
| Guthus | 21 | 19 | 10 | 2 | 2 | 10 | 10 | 12 | 13 | 15 | 15 |
| Sperrebotn | 20 [3] | 15 [3] | 6 | 5 | 6 | 7 | 6 | 10 | 9 | 10 | 9 |
| Støa | nd | nd | 0 | 0 | nd | 20 | 21 | 20 | 21 | 20 | 20 |
| Vaskeberget | nd | nd | nd | nd | nd | 10 | 10 | 10 | 10 | 10 | 10 |
| Huggenes | 28 | 18 | 10 | 12 | 3 | 7 | 9 | 6 | 11 | 17 | 18 |

[1] Part of the agricultural area included in the average values: Augerød 75%, Guthus 100%, Sperrebotn 92%, Støa 70%, Vaskeberget 0%, Huggenes 46%. [2] Only mineral fertilizer was applied. [3] Estimated from Augerød and Guthus (it is expected that the farmers in Sperrebotn showed similar conduct to farmers in Augerød and Guthus).

The increase in P application rates after the mitigation project period (2008–2010) may relate to risk avoidance by the farmers when they are no longer subsidized to reduce their P application rate. The permitted P application rates stipulated in the contracts were lower than the official fertilizer

recommendations, and the extra payment was intended to avoid loss of income from signing the contract. There is also be doubt about the official fertilizer recommendations, in terms of whether they are high enough to sustain high yields. Reijneveld and Oenema [24] found that even well-educated farmers questioned the official fertilizer recommendations. Furthermore, the low price of P supports excessive application [25].

The rate of P application decreased before the start of the mitigation project due to the increased focus on P in the catchment area from 2004. P application rates were also reduced in the period 2008 to 2010, and after the mitigation project, in 2016 to 2017 they remained below the 2004-level, partly because of a reduction in the national P application recommendations in 2007/2008.

In 2005, grass-covered buffer zones (8–12 m wide) were established in Augerød, Guthus, and Sperrebotn along 60%, 100%, and 65% of the streams in these catchments, respectively. From 2008 they covered close to 100% of the stream length running through agricultural land (Figure 2). Støa constitutes only one field, and runoff occurs through the drainage system with no open stream. In Vaskeberget, there was no grassed buffer zone along the short (50 m) stream. Grass covered buffer zones were implemented along 50% of the stream in the Huggenes catchment from 2008 and onwards.

Sedimentation ponds used to trap SS and particle-bound P were established in the stream in Augerød in 2007/08 and in the Guthus and Støa streams in 2009/10. No sedimentation ponds were established from 2010 to 2017.

The action plan implemented for the period 2008 to 2010, in combination with subsidies for no-till, lead to a decrease in autumn ploughing and harrowing. However, the regulation changes implemented in 2013 have influenced the farmers' management and resulted in fewer mitigation measures being implemented from then on.

The mitigation measure package that was part of the contracts with farmers in the Vestre Vansjø catchment targeted both soil and P losses, e.g., no-till in autumn and reduced P fertilization. More farmers producing cereals signed the environmental contracts than those producing vegetables and potatoes. This may be due to the higher economic risk that follows from reduced yields of these crops compared to the economic risk relating to cereal production. The economic surplus (not including work) of onion is approx. 6800 Euro ha$^{-1}$, whereas the economic surplus of barley is 431 Euro ha$^{-1}$ [23]. The high economic value of the crops makes it less attractive to reduce P fertilizer application, which was one of the requirements included in the contract. The establishment of grass-covered areas is correspondingly more expensive for the vegetable farmers compared to the cereal farmers due to the higher income per area of vegetables compared to cereals. Furthermore, the no-till in autumn requirement stipulated in the contracts poses difficulties for producing winter wheat, since most farmers expect higher yields from tillage before winter wheat [13].

### 3.2. Sewage Treatment Renovation

The nutrient contributions to the catchments from sewage are expected to be low, due to low population density in these areas. However, some sewage systems contributed nutrients to the six streams (Table 5).

**Table 5.** Number of sewage systems form single households with direct outlet.

| Catchment | Before 2004 | 2008–2011 | 2011–2017 |
|---|---|---|---|
| Augerød | 8 | 2 | 0 |
| Guthus | 1 | 0 | 0 |
| Sperrebotn | 2 [1] | 2 [1] | 0 [1] |
| Støa | 1 | 0 | 0 |
| Vaskeberget | 1 | 0 | 0 |
| Huggenes | 15 | 0 | 0 |

[1] Pipes from sewage treatment plants was renovated in 2012.

The Augerød catchment contained eight sewage systems from scattered dwellings, and six of these were renovated in 2007 to reduce the nutrient outlet (Table 5). The last two sewage systems were renovated in 2012. In the Guthus catchment, the one sewage systems with direct outlet was renovated in 2007. In the Sperrebotn catchment, the two remaining systems with direct outlets were renovated in 2007. There have also been leakages from sewage pipes within this catchment, and these faulty systems were renovated during 2011–2012.

Before 2004, the Støa, Vaskeberget, and Huggenes catchments had direct outlets from one, one, and 15 sewage systems, respectively. These sewage treatment systems were renovated between 2004 and 2005. Renovating the sewage systems of scattered dwellings can reduce P contributions by 90% [26] and, hence, there has been a large reduction in P outlet from private household sewage systems during the last two decades in the six catchments.

The concentrations of TKB decreased, which may be due to the renovation of sewage systems. The TKB concentrations were below 1600 cfu 100 mL$^{-1}$ in the 24 samples (data not shown) analyzed during 2011–2017. Before 2011, samples with concentrations above this level were identified seven times.

### 3.3. Soil Phosphorus Status

The soil P status was very high (150–200 mg P-AL kg$^{-1}$) in the three catchments with vegetable production (Table 6). The catchment with former chicken production (Guthus) and, thereby, application of P rich manure also has a very high average soil P status (190 mg P-AL kg$^{-1}$). The average P-AL value was lower in the second sampling period (2016–2017) compared to the first sampling (before 2008) in three of the catchments (Table 6), but this decrease was not significant ($p > 0.05$). The average P-AL value is still characterized as high or very high in all six of the catchments (Table 6).

**Table 6.** Average ammonia lactate (P-AL) values (mg P-AL kg$^{-1}$) in each of the catchments, minimum and maximum values in parenthesis ($n$ = 3–7 samples ha$^{-1}$). P-AL values of 0–40 are classified as low, 50–70 as medium, 80–140 as high and >140 as very high [27].

| Catchment | <2008 | 2016–2017 |
|---|---|---|
| Augerød | 110 (30–250) | 90 (30–250) |
| Guthus | 210 (20–390) | 190 (80–360) |
| Sperrebotn | 80 (50–230) | 80 (40–200) |
| Støa | 180 (150–200) | 180 (150–210) |
| Vaskeberget | 160 (110–200) | 150 (110–170) |
| Huggenes | 200 (90–380) | 200 (40–350) |

The reductions in P application rates are expected to contribute to decreased levels of soil P status over time [28]. The rate of decrease in soil P status depends on a priory soil P status, with the largest decline in soil P status seen when the start rate is high [28]. The average soil P statuses of four of the six catchments are above the level at which it is recommended to apply P fertilizer for cereals and grass [27]. It has been shown that reducing P application to zero for very high soil P statuses did not reduce cereal yields [29]. This gives a negative P balance, which over time will result in a reduced soil P status of these soils. However, the recommendation for vegetables is to apply P even when the soil P status is very high. For example, for cabbage, a P application rate of 22 kg P ha$^{-1}$ is recommended for very high soil P status (150–190 mg P-AL kg$^{-1}$) [30]. According to Riley et al. [30], a cabbage yield of 60 tonnes ha$^{-1}$ removes 20 kg P ha$^{-1}$ in the sold product, which gives an approximate balance between P input and output. Soil P status may also be reduced by adding P in balance with the offtake [31,32], but at a lower rate compared to the reduction in soil P status at a negative P balance. Hence, reducing soil P status in areas of cabbage production is slow without risk of decreased yields.

Reduction in soil P status has been shown to be a long-term process [33]. In the Vestre Vansjø catchment, only weak signs of reduction in soil P status were shown during the pilot project period (Table 6). The period here was only 5 to 10 years between soil sampling.

### 3.4. Weather and Runoff

The average annual precipitation in the period was 900 mm (annual min 730–max 1154 mm) (Figure 3). The annual precipitation was slightly higher in the last seven years (967 mm) than the first seven years (835 mm), and there were also more episodes of intense precipitation (>30 mm/day) during the last seven years.

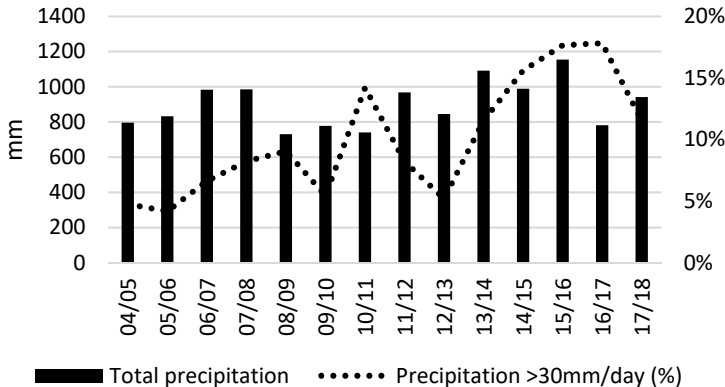

**Figure 3.** Annual precipitation during the monitoring period and precipitation intensities [18].

Mean annual runoff during the monitoring period were 586 mm year$^{-1}$ (min 288–max 658 mm) measured in the Skuterud catchment (approximately 30 km from Vestre Vansjø). During the first year of the monitoring, there were no days with more than 15 mm runoff. The seasonal variation across the study years, except for the summer of 2007, consisted of dry summer months with no runoff above 10 mm day$^{-1}$, followed by increased runoff during autumn and winter.

A regression analysis of the effect of the amount of precipitation and precipitation intensity (Figure 3) showed no significant responses on annual average SS and TP concentrations. Weather and climate constitute complex factors that may override the effect of mitigation measures [10], and the intensity of precipitation is one of the important factors that add to the complexity of these systems [34]. The amount of precipitation each year influences the amounts and sometimes, the concentrations of both SS and TP. The concentrations of SS and TP are correlated to discharge due to the effect of precipitation on the detachment of soil particles and the transport of soil particles and particulate P in surface and subsurface runoff [34]. The effects of mitigation measures may be counteracted by both weather and agricultural land use changes, e.g., intensification of agricultural production [35], and they can also be masked by other processes occurring in the landscape.

### 3.5. Concentrations of Suspended Sediments

The results of water quality analyses showed a general difference in concentration levels between the catchments (Table 7 and Figure 4). In Augerød, Guthus, and Sperrebotn, the highest concentrations of SS and TP were measured in the start of the monitoring period, except one runoff event in Augerød and Sperrebotn on 18 April 2013 when it was raining on frozen soil (Figure 4). Concentrations in samples from Sperrebotn were generally lower compared to the other catchments. Støa, Vaskeberget, and Huggenes gave the highest concentrations in single samples, and this can be seen both in Figure 4, which is limited to samples below 600 mg SS L$^{-1}$ and 1000 µg TP L$^{-1}$ and also in Figure 5 with a full dataset.

**Table 7.** Average annual concentrations of suspended sediments (mg L$^{-1}$) and total phosphorus (μg L$^{-1}$). Range of annual concentrations in brackets.

| Catchment | Suspended Sediments mg L$^{-1}$ | Total Phosphorus μg L$^{-1}$ |
|:---:|:---:|:---:|
| Augerød | 38 (2–490) | 99 (13–547) |
| Guthus | 25 (3–310) | 90 (<3–640) |
| Sperrebotn | 25 (1–230) | 85 (<3–410) |
| Støa | 45 (0.3–1560) | 271 (20–6900) |
| Vaskeberget | 50 (0.3–1780) | 197 (<3–2850) |
| Huggenes | 33 (1–730) | 143 (8–1700) |

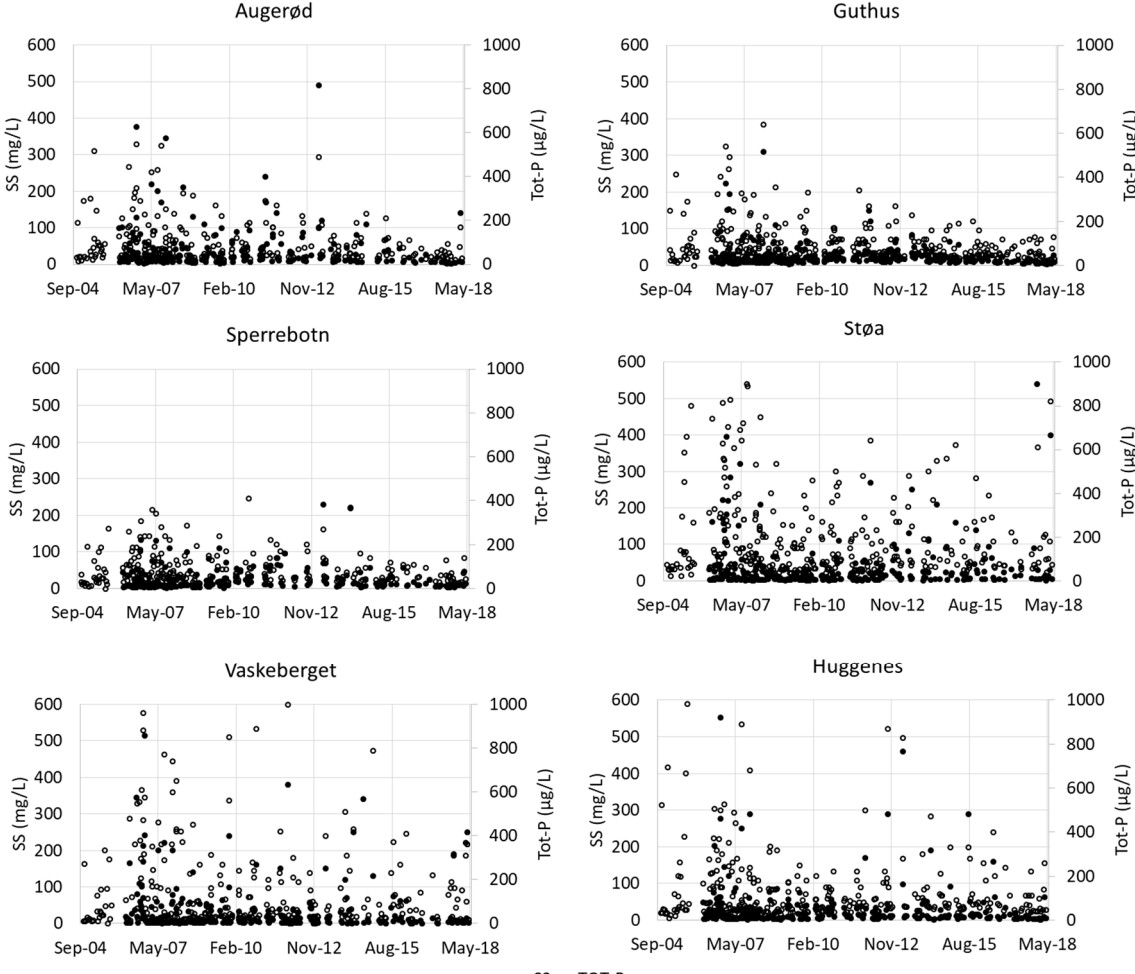

**Figure 4.** Concentrations of suspended sediments (mg L$^{-1}$) and total phosphorus (ug L$^{-1}$) in water samples taken from the six streams during the monitoring period. All samples for Augerød, Guthus, and Sperrebotn and samples below 600 mg SS L$^{-1}$ and 1000 μg TP L$^{-1}$ for Støa, Vaskeberget, and Huggenes.

The highest concentrations of SS (>1000 mg L$^{-1}$) in single samples were measured in Støa and Vaskeberget (Figure 5). The highest concentrations of SS in Støa were 1780 mg L$^{-1}$ in one sample (Figure 5). This occurred on 25 August 2006, after the surface layer of part of the area used for turf production had been removed.

The average concentration of SS for the overall monitoring period varied between 25 mg L$^{-1}$ in Guthus and Sperrebotn and 50 mg L$^{-1}$ in Vaskeberget (Table 7).

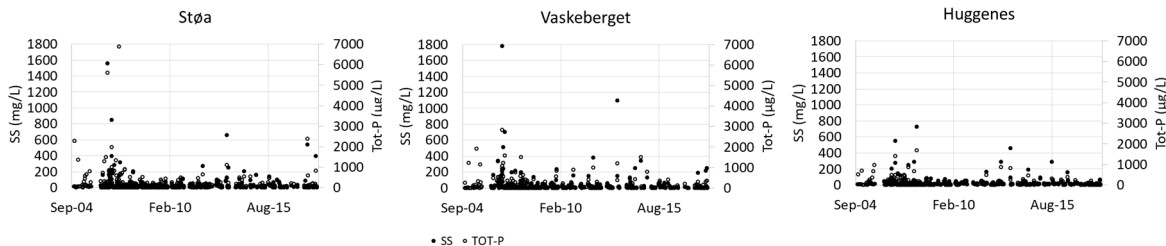

**Figure 5.** Concentrations of suspended sediments (mg L$^{-1}$) and total phosphorus (ug L$^{-1}$) in water samples taken from the six streams during the monitoring period. All samples for Støa, Vaskeberget, and Huggenes.

The annual average concentrations of SS fluctuated during the monitoring period, but there were no significant trends ($p > 0.05$) over time in any of the six catchments (Figure 6). However, there was a tendency towards a decline in annual average concentrations of SS in Augerød and Guthus.

**Figure 6.** Mean annual total phosphorus (µg TP L$^{-1}$) and suspended sediment (mg SS L$^{-1}$) flow-weighted concentrations in the six streams.

The SS concentrations between the six catchments differed, but no single factor explains the difference in average concentrations of SS between the catchments (Figure 7). A regression analyses showed that average annual SS concentrations for catchments were close to significantly affected by percentage agricultural area ($p = 0.09$), but the total catchment area ($p = 0.28$), erosion risk ($p = 0.59$), and autumn soil tillage ($p = 0.17$) were not significant. The SS concentration tended to increase with the percentage of agricultural area (Figure 7). The non-agricultural area comprises a very large part (>80%) of the three catchments, Augerød, Guthus, and Sperrebotn and a minor part (<20%) of the other three catchments, Støa, Vaskeberget, and Huggenes (Table 1). The concentrations of SS in runoff from forested areas are generally lower than concentrations from agricultural land [36], but the concentrations of SS are also influenced by the erosion risk in the agricultural area, and the soil tillage methods used [37].

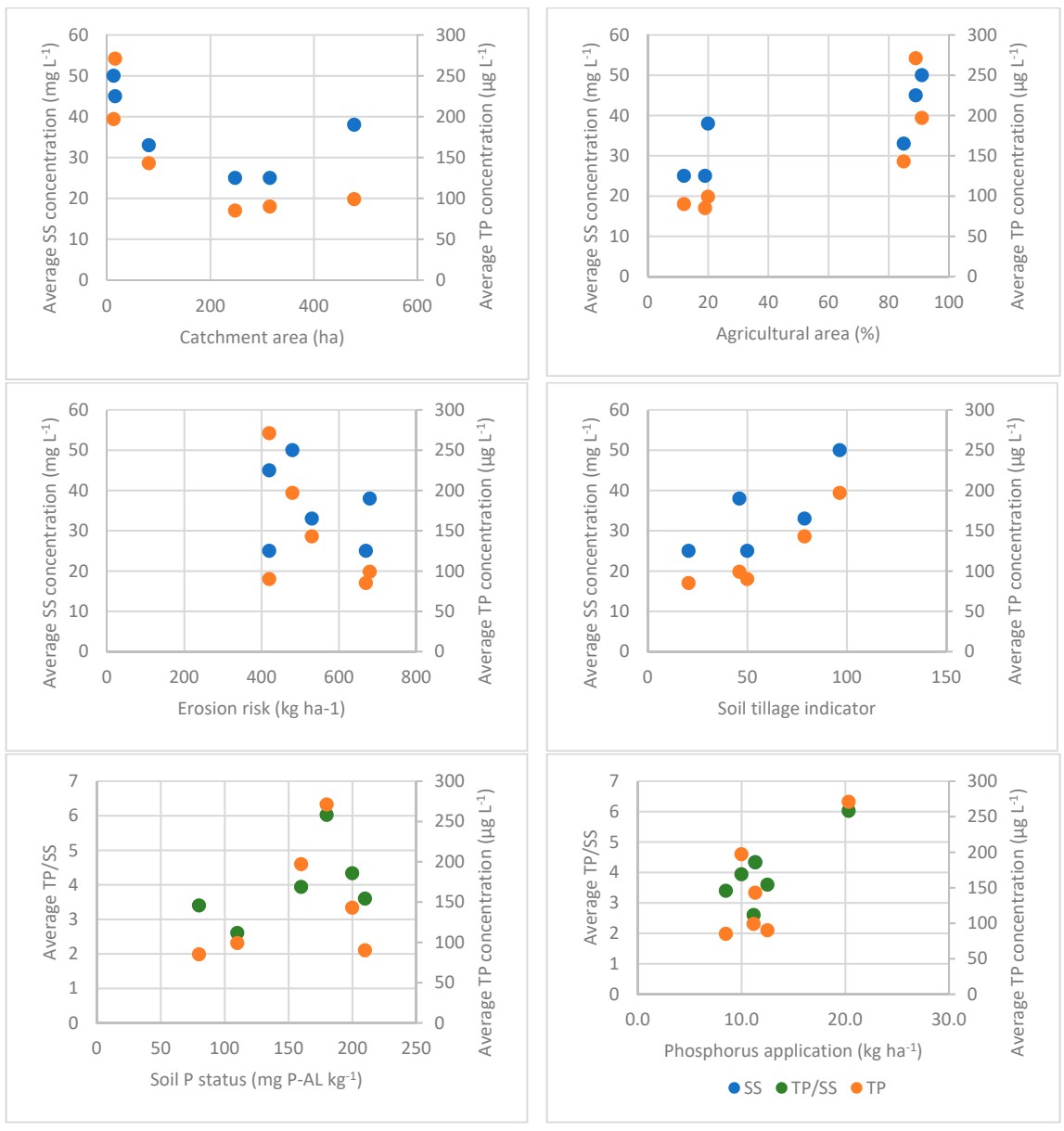

**Figure 7.** Fixed catchment specific effects of catchment area (ha), agricultural area (%), erosion risk (kg ha$^{-1}$) and soil tillage indicator on average concentrations of suspended sediments (mg L$^{-1}$) and total phosphorus (µg L$^{-1}$), and effect of soil P status (mg kg$^{-1}$) and phosphorus application rate on concentrations of total phosphorus (µg L$^{-1}$) and TP/SS.

Average concentrations of SS were highest in the two smallest catchments (Støa: 16 ha; Vaskeberget: 13 ha), and these two catchments also have the highest share of agricultural land (89% and 91%) (Table 1). As a catchment becomes larger, the processes occurring become more complex and these processes may either contribute to increased retention or erosion, e.g., stream bank erosion or furrow erosion [38]. The Støa and Vaskeberget catchments mainly consist of one field each, and the length of the stream is very short with a minimal possibility of soil particle retention occurring, which might explain the high concentrations measured.

The erosion risk in agricultural areas within the six catchments is mainly low (<500 kg ha$^{-1}$) to medium (500–2000 kg ha$^{-1}$), with no areas in erosion risk class 4 (>8000 kg ha$^{-1}$) and only a few percentages falling under erosion risk class 3 (2000–8000 kg ha$^{-1}$) (Table 1). There is no major difference between the catchments, and this factor has no influence on average SS concentrations. The concentration of SS in the six catchments was also in the lower end (25–50 mg L$^{-1}$) compared to the long-term average concentrations measured in agricultural streams elsewhere, e.g., 11–292 mg L$^{-1}$ in Norway [39] and 10–507 mg L$^{-1}$ in Sweden [40].

The average SS concentrations did not significantly increase with average soil tillage in autumn (soil tillage indicator) ($r^2$ = 0.4; $p$ = 0.2). At the plot scale, it is well-known that autumn tillage and bare soil during winter under Norwegian conditions contribute to high concentrations of SS e.g., Reference [6]. This effect is scale dependent [14], but Haygarth et al. [10] showed that mitigation measures like ploughing significantly increased loss of soil particles and colloids also at the catchment scale. The monitoring methods used in that study were much more intensive than the monitoring in the Vestre Vansjø pilot project. However, some indications of such an effect also have been identified. The highest average concentrations of SS were measured in the Vaskeberget, in which the catchment area was dominated by autumn tillage (Figure 2). Furthermore, very little autumn tillage took place in the Sperrebotn catchment, and the concentrations of SS were low even though the agricultural area in this catchment had the highest erosion risk (Figure 2; Table 1). In the Støa catchment, which is flat with a low erosion risk, some high concentrations of SS were found during the first few years after the turf was removed in summer (Figure 4). These single high values of SS concentrations influence the average concentrations for this catchment. In a northern climate with high precipitation during autumn, winter, and spring, and frequent freezing and thawing during winter, open soil during this period contributes to increased risk of erosion [41]. However, at the catchment scale, the effect of mitigation measures, such as no-till in autumn, may be difficult to detect due to the complexity of the natural variability and heterogeneities found in terrestrial and water environments, as discussed by Reference [42]. The effect of changed tillage methods on concentrations of SS is highest in areas with high erosion risk [6], and the changes in concentrations of SS in these areas were expected to be relatively low. However, the predicted increase in precipitation and runoff may cause increased concentrations of suspended sediment and correspondingly increased effects of changed tillage [43].

The agricultural management showed major changes from year to year during the pilot project. A multivariate regression analysis showed annual variables (soil tillage, grassed buffers, sedimentation ponds, precipitation amount, precipitation intensity) did not significantly affect annual average concentrations of SS in any of the catchments. The effect of soil tillage in autumn on annual average SS concentrations for each catchment except Støa is shown in Figure 8.

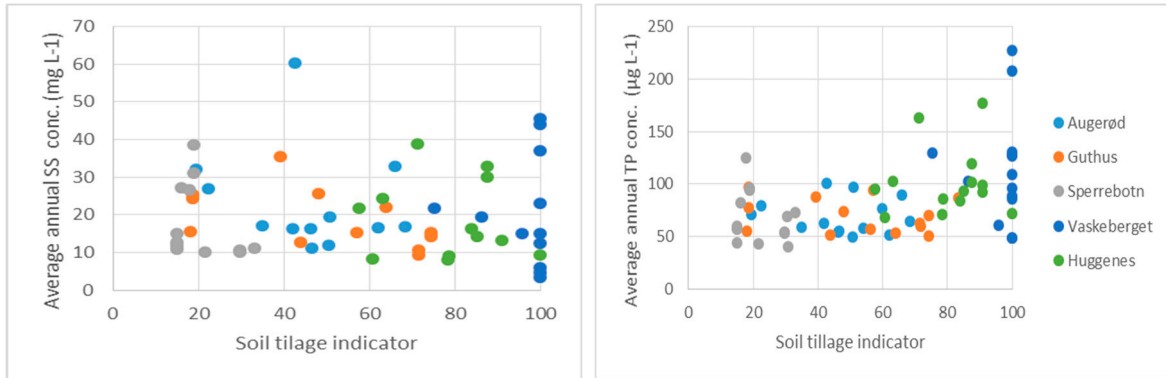

**Figure 8.** Effects of soil tillage in autumn (soil tillage indicator: 100 equals all agricultural area autumn ploughed) on average annual suspended sediment (SS) and total phosphorus (TP) concentrations.

*3.6. Concentrations of Phosphorus*

The average concentration of TP varied between 85 µg $L^{-1}$ in Sperrebotn and 271 µg $L^{-1}$ in Støa (Table 7). The highest concentrations of TP (>1000 µg $L^{-1}$) in single samples were measured in Støa and Vaskeberget (Figure 4). These high concentrations were found in 11 samples from each catchment, nine of which were from 2005 to 2007 (Figure 4). The highest TP concentration was 5600 µg $L^{-1}$. This occurred on 25 August 2006, after the surface layer of part of the area used for turf production had been removed.

The annual average concentrations of TP fluctuated similarly to the SS concentrations during the monitoring period, but there were no significant trends ($p > 0.05$) over time in any of the six catchments (Figure 5). However, there was a tendency towards a decline in concentrations of TP in Augerød and Guthus, and the TP showed a tendency to decrease in Støa, Vaskeberget, and Huggenes.

The relationship between concentrations of TP and concentrations of SS are strong for all of the catchments ($r^2$ = 0.43–0.8) ($p \leq 0.05$). The strength of the relationship decreased in the order from Huggenes, Vaskeberget, Guthus, Augerød, Sperrebotn, and Støa. In Støa, some high TP concentrations were detected after removing the turf. In Sperrebotn, there were contributions from wastewater and individual water samples with very high TP concentrations were found, e.g., 7 September 2010 (Figure 4). Forested areas may also have influenced the relationship between SS and TP for these catchments by contributing SS during specific events. Bechmann et al. [39] showed that, at the catchment scale, concentrations of TP were related to soil P status in addition to concentrations of SS.

The fixed catchment effects of total area, percentage agricultural area, erosion risk, and soil tillage on annual average TP concentrations for the six catchments were investigated (Figure 6). The average TP concentrations showed a significant correlation with the share of agricultural area ($r^2$ = 0.72; 0.03) and this relationship was better than for SS ($r^2$ = 0.56; 0.08). Modelling of effects mitigation measures for all of the Vestre Vansjø catchment using the SWAT-model, showed no significant effect of implemented best management practices when the modelling uncertainty was taken into account [44]. The high proportion of forested area (83%) in that study makes an evaluation of agricultural best management practices difficult.

The multivariate regression analyses showed that annual variables (soil tillage, P application, renovation of sewage systems, grassed buffers, sedimentation ponds, precipitation amount, precipitation intensity) as for SS, did not significantly affect annual average concentrations of TP in any of the catchments. The effect of soil tillage on annual TP concentrations was also not significant (Figure 7). However, as shown in Figure 7 the highest average annual concentrations of TP in individual years were found when and where the soil tillage indicator was high, meaning the high share of autumn tillage.

Concentrations of DRP constitute an average of 20 to 27% of TP for each of the six streams as reported by [16]. For all years, the DRP concentrations were higher in Støa than in the other streams,

especially in 2004/05, when the mean annual DRP concentration in Støa was 110 µg L$^{-1}$. In this year, the DRP concentrations were highest in the three streams draining intensive potato and vegetable production areas. However, since only a few samples were analyzed for concentrations of DRP, conclusions regarding the effect of soil P status and sewage treatment on DRP concentrations were insignificant [16].

*3.7. Sampling Frequency*

The number of water quality samples varied largely between both the individual catchments and years (Table 3). This is due to the size of the streams and the weather conditions. Some of the catchments (Vaskeberget and Støa) are very small, and, therefore, dry out during summer. Other catchments (Augerød and Sperrebotn) are located close to the lake, and for short periods, the lake water ran into the sampling point. No samples were, therefore, taken during these periods. Furthermore, the streams were sometimes frozen during winter and were not sampled at this time. The discrepancies in the number of samples taken each year add to the uncertainty of the estimates of average concentrations. These are factors that mainly relate to sampling in small streams. Also, the number of parameters analyzed varied between the samples. The number of samples taken and the number of parameters analyzed depended on the total budget for each project year. For pilot projects under the auspices of the WFD, the budget is often limited, resulting in low quality of monitoring [15].

Operational monitoring, as it is carried out under the WFD, has been debated in relation to its ability to track changes in water quality over time [11]. The quality of this monitoring is often low since the frequency of sampling is too low to cover the variations in concentrations over time and the effect of mitigation measures [15].

The rate of water quality sampling in this pilot project included sampling at high flow. Concentrations of suspended sediment and TP generally increase with increased flow although this relationship is not unique [45,46]. Therefore, the water discharge when the sample is taken is a very important factor for the concentrations obtained. Time equidistant sampling every two weeks tends not to include the high flow peaks and, hence, not the highest concentrations [47]. In the present pilot project, high flow sampling was, therefore, carried out. However, the high flow sampling was very much dependent on the person taking samples and his/her possibility of foreseeing the flow peak and concentration peak, as well as having the opportunity to do the sampling at the right time. The use of sensors, e.g., for measuring turbidity, would raise the quality of monitoring in this pilot project [48].

## 4. Conclusions

The mitigation project motivated farmers to participate in implementing mitigation measures over the mitigation project period. They were motivated by subsidies offered for implementing the mitigation measures included in the contract, by farm visits, and information campaigns. A reduction in the P application rate of 80% and an increase in the area with no-till in autumn up to 100% of the areas in three of the six catchments were registered.

By the end of the mitigation project period, the farmers withdraw from the contracts. The end of the mitigation project corresponded in time to a change in the general regulation of mitigation measures with a reduction in subsidies for areas with low erosion risk. At that time, P application rates increased again, and a larger area was tilled in autumn. By the end of the monitoring period, however, the P application rates were still lower than it had been before the start of the pilot project in 2004. This decrease in the P application corresponded to a national reduction in P application rate and may not only be due to the mitigation project.

The average soil P status was lower in three of the six catchments by the end of the pilot project compared to the start-values, but this decrease in soil P status was not significant. Changing soil P status is a long-term process.

There were no significant trends in the annual average concentrations of SS and TP over the monitoring period, but the highest single concentrations of SS and TP were detected during the first years of the monitoring. The concentrations of SS and TP were higher in the catchments with a high share of agricultural area and a lower share of contracts, and specifically, the highest single concentrations were detected from these catchments. The lack of significant effects of the mitigation project is due to a short-term (three year) focus on mitigation measures. The low number of water samples taken in certain years and from some certain streams increase the uncertainty and makes it difficult to detect effects. The increase in precipitation during the monitoring period also had an impact on the concentrations.

It is suggested that successful mitigation projects should be long-term, with a consistent focus on mitigation measures. The effects of mitigation measures on water quality are influenced by complex catchment processes and high-quality monitoring together with modelling efforts are, therefore, needed to show the overall effect of mitigation measures.

**Author Contributions:** M.B. was responsible for the consept, methodology, literature review and drafted the manuscript; I.G. was responsible for the calculations and the statistics; A.F.Ø. contributed to Sections 3.1 and 3.3. All authors commented and added to the text; revised the full manuscript and agreed upon conclusions.

**Funding:** The database of this research was funded by the Morsa pilot project, the Norwegian Ministry of Agriculture and Food, and the Norwegian Environment Agency.

**Conflicts of Interest:** The authors declare no conflict of interest.

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
