# Peer review of "Implementation of Mitigation Measures to Reduce Phosphorus Losses: The Vestre Vansjø Pilot Catchment"

_agriculture, doi:10.3390/agriculture9010015_

Round 1
Reviewer 1 Report
General remark
The article is rather a report form application then scientific research. There is a number of references to already published calculations based on this monitoring data. However, the relatively detailed investigation and long term monitoring in the small catchments makes this article interesting. Of course it is a very local case. But, there is not much data from such a small catchments worldwide when the implemented mitigation measures are monitored for such a long time and because of this, it is worth to publish this article after a minor changes.
Detailed remarks
As not every data is available in every catchment it will be more clear if the author will add additional table with information which data is available in particular catchment and the time range.
Row 90 and table. Could you explain more detailed what you understand as erosion risk (%.) and erosion risk classes. May be it is worth to put here the references how the erosion risk was estimated.
Table 2 present data with number of sample in Skuterund catchment. But in table 1 there is no any information about Skuternud. Is this a subcatchment? Please explain the relation between Skuterund and other catchment for example show the catchment border on the figure 1.
Row 142. What do you mean that the runoff is based on Skuterund stream. Could you explain it. In chapter 3.4 are information about runoff. This value is based on Skuterund ? If so it is necessary to show the position of Skuterund in comparison to 6 investigated catchment on figure 1. The information that Skuterund is 25 km from Vansio catchment (row 142) is not enough.
Rows 62 and 78 give the same information
Fig. 2 The areas which are given after the praticular catchment name are the percentage of arable land in the catchment. Could you additionally add to the description of the figure 2 this information. Digits without descriptions is not very clear at the first look. Could you also put the description of the axis’s.
Row 221 and 226. It is not possible to read an average precipitation/runoff form fig. 3.
Row 226. What do you understand the low flow. Could you give a measure of low flow you use. For example probability of occurrences or just explain it.
Row 227. Be more precise when you write "little runoff". It is very subjective.
Row 230. Her you have distance from Skuterund 30 km not 25?
Row 283. Here you first time explain what is a erosion risk class 1. I think it should be explained much earlier.
Row 369. Did you check the statistical significance of this relation?
Reviewer 2 Report
Review comments
The authors describe results and effect of implementation of mitigation measures policies and practices of water quality and P loss in six catchment areas. Data collected before, during and after the implementation of practices and policies. While seeing effect of some practices such as no-till on TSS and P losses at the plot scale, there was limited if any measured impact on the watershed level. Effect of land use and practices on contaminant fluxes and evaluation of mitigation practices efficacy thereof at plot/watershed scales are of great importance. As such this study and results thereof are of interest. However, there is not clear, and sometimes compete absent description of methodology procedure, samples sampling handling and analysis, as well as statistical analysis in the material and methods. This makes it hard, and at points impossible to assess the soundness of the procedure, resulting data, and analysis and interpretation thereof. It is not at all clear what is the punchline of this study.
There are several issues that needs to be address to strengthen the paper for it to be of interest for international audiences, and some that needs to be addressed to provide the clarity and transparency needed before the paper is ready for publication.
The main concern is the lack of in-depth analysis of the data and statistical analysis thereof. Namely, parsing the dataset by before/after, and any fixed/random variable/effect across time/landscape, assessing random and fixed effects to push this study from being a local anecdotal to one that can more categorically assess the treatment effect (any/one of the programs) on any one of the measured variables. Data needs to be pooled by the different variables (%ag land, %ag practice intensity, rain intensity, landscape morphology, % land of participants, etc.) to assess the impact of the practice(s) on the measured parameters. The regression analysis that is alluded to in the discussion needs to be explained in the M&M and presented in the Results section. Trying to explain the observed response in reference to any of such variable will strengthen the paper to become worthy of reading by audience with interest outside the tested areas. Also, the analysis needs to be done on the pool of data before (during) and after the implementation of any of the different programs and policies. Analysis thereof needs to be done on the plot level
Overall, the paper needs to: include proper introduction of the problem/topic (in both abstract and body – the authors jumping to the program – WFD – without providing the justification and background – answering the ‘why’ question before going to haw we remediate a problem that you yet to introduce…); provide detailed description of material and methods used with a refine statistical analysis; provide robust synthesis impact and implications of the results – that may hold beyond the tested area; provide specific, bullet-type statements, summarizing the findings, outcomes, and any recommendations.
For the paper to be of value to regional and international audience (beyond the pool of interested parties in the locality of it) the analysis of the results needs to depart from being watershed- specific (anecdotal) to be associated with more ‘universal’ factors that may affect the observed behavior/trends/changes of the measured variables. And that one can interpolate from this study to other locations/situation. This may include using such universal factors as co-factors or parameters in regression analysis. For example, try to find correlation between weighted-percent land use (ag/forest, etc.) and measured variables; % ag practice (intensive/extensive) and compliance or impact on measured parameters. If not enough data exist, try to use pair comparison (t-tests) for before/after/during time type comparison. The authors mentioned for example that farmers practicing intensive agriculture were less incline to participate in the P reduction program – is this just an anecdotal observation or can that be correlated across watersheds and impact on plot/watershed level (i.e. % of participants per ag cropping system, etc.). Other factors may be landscape morphology (e.g. %slope/%area, rain/in-stream flow intensity, erosion risk, etc. The authors discuss (in the discussion section) some correlations among such parameters (e.g. TSS and TP). This are very important and display of them in graph/table is of great value to the readers. The results need to be presented in the result section – as figures with the regression line coefficient and goodness of fit, or in tabulated forms (e.g. Pearson correlation, stepwise regression analysis, etc.), and/or in ANOVA table/tests with well define (main) treatment effects.
Below are some specific comments:
Abstract – introduce the problem (e.g. ag field and practices as non-point source for P and sediment loss).
Introduction
Here again, the topics are in invers order - introduce the problem and then, WFD as (one?) of Norway approaches to mitigate that problem.
L30 change huge to large
L42 add ‘water’ bodies after “surface”
L58 it is not clear if the name of the lake is “Western Vansijo” (note the capital letters), or if it is a lake located western to Lake Vansio…
Material and methods
Trying to understand how many programs were implemented and duration thereof is very confusing. I strongly suggest the authors to provide a visual diagram with timeline of start/end of each of the different programs (e.g. WFD pilot 2004 - ?; Pilot mitigation by ministry of Ag, 2008-2010; county governor 2011 – 2013, general regulation 2009-2012 [L279], change in no-till regulation [2013-on, L281] etc.). Consider including sampling/monitoring start/end date for each of the measured variables – see how crowded that turn to be and maybe split into two diagrams. Currently it is very confusing to try to understand what program was implemented and when, what exactly each of the 4-5(?) programs were for and what was implemented in each (suggest tabulate this information). Nor is it clear what was measured and when. Having the timeline in a visual diagram will be very helpful and having the different programs, their timelines, and practices in a tabulated manner will add clarity.
Table 1. explain or reference what erosion risk is and the different classes.
L68 ‘normal’ – average?
L68. What is “DNMI”?
L69 replace “are” with ‘range’
Fig 1 – not clear where is the lake in this plate
Provide rational/explanation why these watersheds were chosen.
Table 2 – Skuterud catchment is not defined/introduced/discussed anywhere in the paper. As such, what this information is for? Why providing it to one area that is not included in the study, and provide none to the ones that do?
How was runoff measured?
Not clear what are the headings (e.g. “04/05” year?) need to clearly specified and labeled.
L139 – is “Eurofins” soil testing lab? – clarify; “ammonia lactate extraction method” – briefly describe the method or provide English reference thereof.
Section 2.6 – here again, Skuterud is not in the map or table 2. Not clear why it is included and if it was at all under any of the programs.
L145 change “table” to level; change “in a” to using; if pressure transducer was used to monitor water level provide make and model; add Scientific after Campbell; provide specs of datalogger.
L149 add analysis after regression and test after Mann-Kendall
Was any other statistical analysis done, on any of the other measured variables? What was the (main) ‘treatment effect” – the before and after? Was any additional parameter tested to try to explain the results (share of ag area, share of ag crop/practice, etc.)? How were fixed effects treated? How the unbalanced dataset was statistically treated?
Results
L154 – change 20010 to 2010.
L167 change yearly to annual
L175 – need to clarify what “open” stream means.
Fig 2 Label the x-axis – year?
What is “gras’in legend - is this “grassed waterway” (L100), “grassed buffer strips” (L99)?
If any one of these, then the y-axis is not clear - % of what? The fields or the stream? Meaning, how do one present a situation where for example 100% of the agronomic lands are ploughed while 100% of the stream banks has buffer strips?
L193-194 – clarify what “renovated” and “improved” means.
Table 4. – title – consider adding ‘to stream’ after outlet
L199 replace leaching with leaking
L221-224 – provide reference or data to support the statement (“… 90%...”)
L205 – “showed” – where? Provide data or reference. Explain what “TKB” is.
Soil P status – where were these soil samples collected (ag field, stream bank, etc.)? What was the sampling scheme/grid, depth, etc. Need to describe the sampling methods and protocol as well as where it was collected from in the material and method section.
Table 5 - Explain why the these time periods were chosen. if the mitigation practice are from 2008-10, why grouping 2007 with them. Why not comparing the before (any data from 2004 through 3007) to after? Add standard deviation to the averages. Consider (here and throughout the paper) the site order in the table – alphabetic or by some of the major variable content (e.g. ascending order of abg P-AL)
L220 – reevaluate the range of temperature provided and how, if at all, meaningful is using such average – is the temperature in Norway fluctuate between 4 to 10 C…?
L221 annual average precipitation?
L221, L226 reference to “Figure 4” and “Figure 3” is wrong, consider changing figures numbers to conform to their reference in the text.
Figure 3. very crowded and not helpful at all. Also, and again where is this site and how it is relevant to the study – add it to Table 1 and Fig 1 and explain the ag management/participation in any of the said programs.
Table 6 (and L237-8): average of what? All the data in one year? All years? If the latter (as suggested in L237-8) then what is the rational of doing so – pooling data from before/during/after the programs?
Figure 5 – hard to read and asses – data too compressed on the Y-axis. Consider changing the Y-axis scale and/or to introduce breaks therein.
Figure 6 – flow-weighted concentration – this concept and calculation needs to be introduced and explained in the material and method section
L283 – add “/y” to the erosion rate.
L277-286 – try to pool and compare before and after (no-till periods – group as in fig 2; or by site as proxy thereof)
L297 – main project period – change to 2008-2010 (see L92)
Soil P status (L288 – on) – try comparing the data by ag crop/practice intensity by year or by period – (before and after mitigation project)
L347 “open soil”? bare soil maybe?
Reviewer 3 Report
The objective of this study was to evaluate a pilot project for implementation of mitigation measures to reduce phosphorus (P) losses. This is an interesting study, and the results could be important not only for researchers but also for the planners, local governments and farmers as well for catchment managers. Data presented in the study covered a 14-year period, and even if it was not enough to obtain clear correlations between measured data and measures implemented, in my opinion this paper is worth to be published.
My remarks are as follows:
1. Introduction – line 50 – I suggest to describe (base on the literature) possible effects (focused on SS and P) of different measures introduced in pilot catchments
2. Percentage of the catchment area covered by the farms included in the project could be add to table 1
3. Line 118 – 1st of May – to 30th of April?
4. Line 122 – delete “respectively”
5. Table 2 – please describe the “Skuterud catchment” and explain the reason why it is used here
6. Results: lines 152-157 – can be shifted to “materials”
7. Line 164: please describe the type of the fertilizer (mineral/organic?)
8. Line 174 – I could not read it from the fig.2
9. Fig 2 – I don’t understand it. Please describe how the % was calculated? Is the total equal to 100%? Please rethink
10. Page 7, lines from 192 and table 4: this part is not clear. I also don’t understand the index “2” – why it is only in one place if the renovation of pipes was done in many cases?
11. Lines 205-207 it was not included in “methods”
12. Table 5: please add reference to the limits in the table description
13. Fig. 5 – please improve the quality
14. Fig. 6 – I suggest to use the loads (not concentration). From the reader perspective it would be very interesting to know the loads in kg/ha/yr
15. Line 278-284 – should be in project description
16. Is there a correlation between soil P status in catchment and P in streams?
Round 2
Reviewer 2 Report
Review Comments 2nd round
Inasmuch as progress been made there is still significant work to be done to ‘bring this product to market’. This relate to clarity as well as concise conclusion(s), outcome(s) and impact of the study.
It will really be appreciated if the authors will dedicate more time improving the flow and larity of the study and provide clear conclusive remarks to summarize it and its potential impact on the subject matter (either the validity of the practices, or the study in combating SS and TP input from ag lands).
Inasmuch as I do not address each and every place where change and/or additional/clarity needs to be added/made, the authors should apply healthy level of scrutiny of their product to bring it to where it needs to be.
For any future reference - a ‘clean’ version needs to be provided – the last version clean from any revisions.
More specific comments:
Title: change ‘implementation’ to Effect
L2. Change ‘watercourse’ to waterbodies
L17. Add ‘of’ after ‘comprised’
L18. Add ‘during this period’ after ‘decrease’
L18-19. ‘decrease’ and ‘increase’ – provide hard values - % decrease/increase.
L42 change further to furthermore.
L104-107 – very long and confusing sentence.
L106 consider replacing ‘and’ with ‘or’ and ‘but’ with ‘and’
L109 add ‘in Norway’ after ‘initiated’
L111 groundwater (one word)
L113 replace ‘involvement’ with ‘participation’
L117-122 – this should be one paragraph (not two).
L119 delete ‘therefore’
L119 replace ‘evaluate’ with summarizes
L129. Add ‘annual’ before ‘precipitation’
L194 ‘placed limits on…’ – provide details – what limits.
Table 2. define “ER”
L331. If correct, add ‘which include all these watersheds’ after ‘county governor of Ostfold’
Please note – this gap in line-item input does not mean that all is well between L331 and L752.
L752. Consider replacing ‘will’ with did
L752. Delet ‘to’
L754 replace where with when
L757-760 – long and unclear sentence.
L762 – replace “could be” with were.
There still is a need for clarity about the treatments (policy/practice implemented; “time effect” to some extent), use of consistent nomenclature/definition thereof throughout the text, and synchronization thereof with the timelines. All are still confusing.
The added sketch for the project/timeline combination is very helpful but there are still some confusion given the non-consistent use of terminology for the different periods and the related duration thereof – see more details below.
The term ‘project period’ is used in different versions multiple times throughout the text (‘the project period’ L140; L332; L1426; ‘the main project period’ L218; ‘main pilot project period’ L450, etc.). It is not clear to which project the authors refers. For the sake of clarity, either define the different project periods differently and stick to that definition throughout the test, or better yet, provide the relevant years when referring to ‘a’ project – as in L522 (‘the main project period (2008-2010)’).
‘project period’ used and referred to in Table 6 and related text, in L762, and lines L422-425 – the authors pool 2007-2011 data to calculate average P-AL. Given the main project periods (2008-2010), it is not at all clear what a 2007-2011 average values actually represents as they cover pre (2007) and post (2011) main-project (2008-2010) periods in addition to the main project period. The authors need to provide a rational explanation for their decision to pool the data from unrelated periods and to define what the average results represent – is it the average P-AL for the main-project period? If so why including the 2007 and 2011 data? If not, then what type of treatment (i.e. application of any policy/practice) does the average values for the 2007-2011 period represents? All in all, some consistency in definitions for the different period and use thereof throughout the text, as well synchronization between the treatments/terms (any ‘project period’) and the timeline needs to be conducted or otherwise explained.
Table 3 and runoff. It is not clear what the data on number of samples for TP and SS in table 3 are for – is this for Skuterud? Is it for ALL the catchments used in this study, for selected ones? The authors need to provide information on that.
L358 ‘By linear interpolation, annual loads were estimated and flow-weighted concentrations calculated from the annual loads divided by annual flow’. Annual loads estimated FROM WHAT? What is it that was used from the Skuterud watershed to estimate annual loads – what was interpolate? I strongly suggest rethink and rewrite this section. It is not clear what was used, how it was used, and to what ends.
The authors state that data from Skuterud was used to calculate annual loads and flow-weighted concentrations. Where are the calculated “annual loads” and “flow-weighted concentrations” reported? I may be missing this.
L769 ‘Mean annual runoff ‘ – if I understand this correctly, these values are ESTIMATED using the Skuterud runoff data. If so, add ‘Estimated’ before ‘Mean…’ If this is data from Skuterud that state it as such.
Figure 4. is useful for showing temporal changes – i.e. any trend/change that occur overtime, which there is little if any to show for, or distinguish in such compressed figure where only some outliers jump out (the authors do a good job in explaining the off-trend spikes in L863 for one of the sites/land use - if so, how about the outliers in the other locations?). If the authors identify any temporal change in any of these variables they should construct the figure in a way that it will show it. If there isn’t any trend, this should point to as well when discussing this figure.
L1086 and Figure 6. Use the entire and raw dataset (rather than overall average), in the figure. If only to overcome the momentum bias inflicted by the high/outlier value for one of the sites as stated in L1096: “These single high values of SS concentrations influence the average concentrations for this catchment.” This is exactly the point for using the raw dataset rather than any descriptive statistic in such correlation (e.g. use the dataset used to construct Fig.4).
Figure 7. “Time dependent effects”. Currently the figure shows soil tillage indicator effect, not time – also see related discussion in the text.
Correlation analysis (in general) – the authors discuss correlation analysis (e.g. L775, L872, L1103, L1299) but do not provide results – either in tabulated form of in graphs. You should present the results and analysis to be able to discuss them.
Author Response
Thank you for very good comments. We have responded to Your comments here below and have revised the manuscript. Due to a strict deadline for the revised manuscript there was no time to change the dataanalyses in Figure 7 and use raw data there. I hope you will get a clean Version of the manuscript to be able to read it.
Comments and answers to Reviewer 2 Second round
Once again we want to thank the reviewer for giving very precise, appropriate and helpful comments. We see now that the suggested changes were really needed and we believe that the manuscript has improved a lot due to the suggested additional analyses
Inasmuch as progress been made there is still significant work to be done to ‘bring this product to market’. This relate to clarity as well as concise conclusion(s), outcome(s) and impact of the study.
It will really be appreciated if the authors will dedicate more time improving the flow and clarity of the study and provide clear conclusive remarks to summarize it and its potential impact on the subject matter (either the validity of the practices, or the study in combating SS and TP input from ag lands).
We have improved the conclusions
Inasmuch as I do not address each and every place where change and/or additional/clarity needs to be added/made, the authors should apply healthy level of scrutiny of their product to bring it to where it needs to be.
We have revised the whole manuscript to clarify the content
For any future reference - a ‘clean’ version needs to be provided – the last version clean from any revisions.
A clean version of the manuscript has been provided, it’s up to the editor to forward a clean manuscript
More specific comments:
Title: change ‘implementation’ to Effect
The manuscript describes a pilot project with the main aim of implementing mitigation measures for improved water quality. The effect of the project on implementation of measures is one of the objectives and therefore the authors think that “Implementation” better covers the content than does “Effect”.
L2. Change ‘watercourse’ to waterbodies
Has been changed
L17. Add ‘of’ after ‘comprised’
This has been changed. The manuscript was edited by a professional language editor, but they did not mention this.
L18. Add ‘during this period’ after ‘decrease’
Has been added
L18-19. ‘decrease’ and ‘increase’ – provide hard values - % decrease/increase.
Numbers has been added
L42 change further to furthermore.
Has been changed
L104-107 – very long and confusing sentence.
The sentence has been changed
L106 consider replacing ‘and’ with ‘or’ and ‘but’ with ‘and’
Has been changed
L109 add ‘in Norway’ after ‘initiated’
Has been added
L111 groundwater (one word)
Has been changed
L113 replace ‘involvement’ with ‘participation’
Has been changed
L117-122 – this should be one paragraph (not two).
Has been changed
L119 delete ‘therefore’
Has been deleted
L119 replace ‘evaluate’ with summarizes
Has been changed
L129. Add ‘annual’ before ‘precipitation’
Has been added
L194 ‘placed limits on…’ – provide details – what limits.
Has been added
Table 2. define “ER”
Has been included
L331. If correct, add ‘which include all these watersheds’ after ‘county governor of Ostfold’
The sentence has been changed
Please note – this gap in line-item input does not mean that all is well between L331 and L752.
We have read the text thoroughly and revised
L752. Consider replacing ‘will’ with did
Has been changed
L752. Delet ‘to’
Has been deleted
L754 replace where with when
Has been changed
L757-760 – long and unclear sentence.
The sentence has been improved
L762 – replace “could be” with were.
Has been changed
There still is a need for clarity about the treatments (policy/practice implemented; “time effect” to some extent), use of consistent nomenclature/definition thereof throughout the text, and synchronization thereof with the timelines. All are still confusing.
The added sketch for the project/timeline combination is very helpful but there are still some confusion given the non-consistent use of terminology for the different periods and the related duration thereof – see more details below.
The term ‘project period’ is used in different versions multiple times throughout the text (‘the project period’ L140; L332; L1426; ‘the main project period’ L218; ‘main pilot project period’ L450, etc.). It is not clear to which project the authors refers. For the sake of clarity, either define the different project periods differently and stick to that definition throughout the test, or better yet, provide the relevant years when referring to ‘a’ project – as in L522 (‘the main project period (2008-2010)’).
The timing of project periods and regulations has been improved to clarify the timing throughout the text. There are two different levels of projects: The pilot project (2004-2017) and the mitigation project in Vestre Vansjø (2008-2010). This has been defined in the text.
‘project period’ used and referred to in Table 6 and related text, in L762, and lines L422-425 – the authors pool 2007-2011 data to calculate average P-AL. Given the main project periods (2008-2010), it is not at all clear what a 2007-2011 average values actually represents as they cover pre (2007) and post (2011) main-project (2008-2010) periods in addition to the main project period. The authors need to provide a rational explanation for their decision to pool the data from unrelated periods and to define what the average results represent – is it the average P-AL for the main-project period? If so why including the 2007 and 2011 data? If not, then what type of treatment (i.e. application of any policy/practice) does the average values for the 2007-2011 period represents? All in all, some consistency in definitions for the different period and use thereof throughout the text, as well synchronization between the treatments/terms (any ‘project period’) and the timeline needs to be conducted or otherwise explained.
We have chacked these data and found that there was in fact no P-AL-values from 2009-2011, but some samples were from early 2000, therefore the text has been changed to “Before 2008”.
Table 3 and runoff. It is not clear what the data on number of samples for TP and SS in table 3 are for – is this for Skuterud? Is it for ALL the catchments used in this study, for selected ones? The authors need to provide information on that.
Table 3: The Table text has been improved and the content explained better
L358 ‘By linear interpolation, annual loads were estimated and flow-weighted concentrations calculated from the annual loads divided by annual flow’. Annual loads estimated FROM WHAT? What is it that was used from the Skuterud watershed to estimate annual loads – what was interpolate? I strongly suggest rethink and rewrite this section. It is not clear what was used, how it was used, and to what ends.
The authors state that data from Skuterud was used to calculate annual loads and flow-weighted concentrations. Where are the calculated “annual loads” and “flow-weighted concentrations” reported? I may be missing this.
Calculation of the annual flow-weighted concentrations has been described in detail
L769 ‘Mean annual runoff ‘ – if I understand this correctly, these values are ESTIMATED using the Skuterud runoff data. If so, add ‘Estimated’ before ‘Mean…’ If this is data from Skuterud that state it as such.
Data on runoff is measured in the Skuterud stream. This has been specified
Figure 4. is useful for showing temporal changes – i.e. any trend/change that occur overtime, which there is little if any to show for, or distinguish in such compressed figure where only some outliers jump out (the authors do a good job in explaining the off-trend spikes in L863 for one of the sites/land use - if so, how about the outliers in the other locations?). If the authors identify any temporal change in any of these variables they should construct the figure in a way that it will show it. If there isn’t any trend, this should point to as well when discussing this figure.
The Figure has been changed to a different scale to better show the variation between catchments and another Figure has been added to show the full dataset of the catchments with the highest concentrations. Comments on ourliers has been added toi the text.
L1086 and Figure 6. Use the entire and raw dataset (rather than overall average), in the figure. If only to overcome the momentum bias inflicted by the high/outlier value for one of the sites as stated in L1096: “These single high values of SS concentrations influence the average concentrations for this catchment.” This is exactly the point for using the raw dataset rather than any descriptive statistic in such correlation (e.g. use the dataset used to construct Fig.4).
Due to the tight deadline for the revised manuscript it was not possible to do a new dataanalyses including the full dataset. It would have been nice to do this, but at the same time such an analyses would be very much influenced by the extreme values found in single samples.
Figure 7. “Time dependent effects”. Currently the figure shows soil tillage indicator effect, not time – also see related discussion in the text.
You are right; it shows the effect of tillage, which changes from year to year. The text has been changed.
Correlation analysis (in general) – the authors discuss correlation analysis (e.g. L775, L872, L1103, L1299) but do not provide results – either in tabulated form of in graphs. You should present the results and analysis to be able to discuss them.
Results from the correlation analyses has been written in the text with r-square and p-value.